# Toward sustainable community-based tourism development: Perspectives from local people in Nhon Ly coastal community, Binh Dinh province, Vietnam

Tuyen Dai Quang[1], Quynh Xuan Thi Nguyen[1], Hoang Van Nguyen[2]*, Vang Quang Dang[3], Nhan Thong Tang[2]

1 Faculty of Tourism, Nguyen Tat Thanh University, Ho Chi Minh City, Vietnam, 2 Faculty of Tourism and Hospitality Management, HUTECH University, Ho Chi Minh City, Vietnam, 3 Faculty of Economics – Ho Chi Minh City University of Technology and Education, Ho Chi Minh City, Vietnam

* nv.hoang79@hutech.edu.vn

## Abstract

Community-based tourism (CBT) is no longer a new form in many countries, including Vietnam. However, the approach of CBT in a sustainable manner, especially in coastal villages, where tourism has recently emerged as an alternative livelihood, has received little attention. This study explores the impact of tourism on the lives of the local community, the level of people's participation in tourism, and the challenges of tourism development in the coastal commune of Nhon Ly, Binh Dinh province. By utilizing a qualitative method such as observation and semi-structured interviews with 19 villagers engaged in tourism, the results show that tourism contributes significantly to the lives of the residents in Nhon Ly (e.g., improving income, changing livelihoods from fishing to tourism, improving environmental awareness). However, tourism development has obstacles related to local people's skills and knowledge working in tourism and their limited participation in tourism policy and planning. Other issues were related to the negative impacts of tourism on cultural-social and environmental aspects in Nhon Ly. Therefore, there is a need for comprehensive coordination among all tourism stakeholders, building long-term strategies to overcome barriers toward sustainable CBT development in the future.

**Data Availability Statement:** All relevant data are within the paper.

**Funding:** The authors received no specific funding for this work.

## 1. Introduction

Today, tourism is one of the most growing industries. Tourism plays a vital role in fostering a nation's economy worldwide. It enhances opportunities, allowing policymakers with the skills and knowledge to formulate policies and effective tourism development strategies. Tourism can lead to the destination's competitiveness and sustainability while benefiting local communities. Communities are the primary focus of goals as they play an essential role in the sustainable destination development [1, 2].

Coastal tourism has been developed for a long time, known as the "3S tourism model" (Sun, Sea, and Sand) [3]. However, tourism in the coastal areas is a complicated phenomenon

**Competing interests:** The authors have declared that no competing interests exist.

because it brings both positive and negative impacts on the environment, economics, and socio-culture. Hunter and Green [4] found that tourism in coastal areas leads to adverse environmental effects such as erosion from overuse of beaches, increased water pollution due to poor wastewater management, and loss of marine and coastal habitat [5]. Consequently, it affects tourists' decisions [6, 7]. Social and cultural activities are another major attraction for tourists but can also be irreversibly ruined by the impact of tourism. King & Stewart [8] argues that the increase in tourists and the power of money has turned the local environment and culture into goods. An integrated approach can be transformed into sustainable coastal tourism development to conserve and develop local destinations. At the same time, the various strategies, development plans, and policies need to be thoroughly researched before projects are implemented and monitored after implementation [9, 10].

The adverse problems occurring due to the proliferation of conventional mass tourism activities have increased the interest of researchers and decision-makers. They often focus on new forms of sustainable tourism as it involves a lot of awareness, planning, local control, and conservation and development opportunities [11, 12]. They argue that sustainable coastal tourism development and management with integrated measures can help resolve imminent conflicts of social needs. Properly implemented sustainable tourism development can provide long-term opportunities for long-term coastal tourism and enable effective environmental protection policies [5, 13].

Vietnam is a country with territorial waters and a continental shelf under national sovereignty and jurisdiction, nearly 1 million square kilometer wide (3 times larger than the land area) in the East Sea (Biển Đông). It has a coastline of 3,260 kilometer and about 3,000 large and small islands near the shore and two archipelagos of Hoang Sa and Truong Sa, with 28 provinces and cities bordering the sea, accounting for 42% of the land area and 45% of the country's population [14]. The coastal islands developed for CBT include Bach Long Vi, Ha Long, Cat Ba, Con Co, Cu Lao Cham, Ly Son, Con Dao, and Phu Quoc. The coastal fishing villages have many favorable conditions for developing community-based tourism, mainly in the Central region, from Quang Binh to Binh Thuan provinces.

Considered a rich tourism potential, Binh Dinh is gradually becoming a destination attracting the attention of both domestic and foreign tourists. The province has implemented policies to promote the potential of natural and cultural resources in coastal areas. Over the years, many investment projects have been approved to develop coastal areas, marine tourism, and ecotourism services throughout the coastal regions. Recently, CBT projects have been implemented in localities of the province, such as Quy Nhon, Nhon Hai, Nhon Ly, and Nhon Chau. However, the development of CBT in Binh Dinh is still not commensurate with the province's potential, especially in the coastal area.

Moreover, the development of CBT in these localities has not brought economic benefits to the local community. This development requires research to identify the issues of sustainable CBT development in the locality. Therefore, this study explores the following three questions: (1). How has tourism impacted on socioeconomic of the community in Nhon Ly? (2). What are the forms and levels of participation of the local community in tourism activities in Nhon Ly? (3). What problems exist in tourism today and what are their causes?

While many coastal destinations are facing significant issues of environmental impacts, our case study shows that the Nhon Ly community is intensely aware of the importance of protecting marine resources to serve their community's tourism. This study affirms the importance of community consciousness and self-governing community models to ensure sustainable tourism development for tourism destinations. This paper is structured as follows: Section 2 reviews the literature, including coastal tourism, sustainable tourism, and CBT, section 3

describes the methods, section 4 represents the research context, section 5 discusses the findings, and followed by the discussion and conclusion in the last section.

## 2. Literature review

### Coastal tourism

Coastal tourism involves local communities, visitors, and tourist destinations in a coastal environment and its natural and cultural resources [15]. It is a particular type of tourism due to the isolation, diversity of ecosystems, and specificity of sites, affecting the characteristics of residents [16]. Most coastal tourism occurs along the coast and in the water off the shoreline. Coastal tourism activities take place outdoors and indoors in sports and leisure, as well as leisure and business [17]. In addition, the unique ecology and folklore make coastal destinations more attractive and exciting [18]. Coastal areas have become one of the most popular tourist destinations [19]. According to Royle [16], as coastal areas and islands often have limited ecological diversity and different characteristics, most islands have unique flora and fauna, exotic coral reefs, and marine life, creating each fascinating surroundings and landscape. Therefore, it attracts many tourists to experience [20, 21].

Many studies reveal that tourism in the coastal and islands is a double-edged sword that affects the local community [22, 23]. Like other human efforts in coastal areas associated with development, tourism is viewed positively by some researchers for the opportunities it creates [24]. Others claimed that tourism in sensitive areas negatively impacted the local coastal destinations [23, 25]. For example, in Ismail and Turner's [26] study, tourism stimulates the development of basic infrastructure, contributes to the growth of local industries, and attracts foreign investment. In addition, environmental conservation and improvement are promoted, and regional diversity and culture are preserved. Therefore, tourism significantly influences destinations on ecological, economic, and socio-cultural issues [13]. However, due to the limited capacity of the coastal area and island in environmental, social, and economic aspects, these places develop unsuitably as tourism activities rely entirely on natural resources. It leads to severe environmental and social imbalances and destination economic specialization [25, 27].

Furthermore, Shaw and Williams [21] argue that although tourism has a positive impact on coastal areas, it has the potential to hurt local communities. Similarly, Ap [28] also claims that tourism is vital in stimulating economic growth and sustaining development in the coastal area and small islands. Still, the rapid increase in tourist arrivals has put pressure on tourism and forced the local community to participate in that development. According to Macleod [29], although tourism is a more critical factor in an island's economy, it also impacts the community's socio-cultural structure because of its small size, and socio-cultural effects are higher through regular contact between local people and tourists.

In addition, the negative issues of coastal tourism are related to geographical and environmental features. Dodds, Graci, and Holmes [25] argue that while local people benefit from improved economic and well-being from tourism, they also suffer damaging environmental and social consequences. Such negative issues include environmental overgrowth and the dependence of the local community economy on tourism. In addition, the negative also comes from overuse of natural resources, congestion of infrastructure, leading to pollution and resource degradation and elimination of the natural environment, along with competition and pressure on the destination infrastructure [30, 31].

### Sustainable tourism

According to The World Tourism Organization (UNWTO), sustainable tourism is defined as "tourism that takes full account of its current and future economic, social and environmental

impacts, addressing the needs of visitors, the industry, the environment, and host communities." Sustainable tourism development is required three crucial foundations of economic, socio-cultural, and environmental sustainability [32–34]. In addition, sustainable tourism is achieved if a destination can accommodate the number of visitors at a given time without damaging the aspects of economics, culture, and environment [35].

Presently, many coastal destinations are adopting solutions for sustainable tourism development. However, there is no standard formula for managing tourism's negative and positive environmental impacts as some environments can be much more unique, fragile, finite, and irreversible than other environments [25, 36]. Second, environmental impacts caused by tourism establishments are often difficult to distinguish from other factors, so one effect may cause another [18, 20]. Sustainable tourism development is an ongoing process, and not a specific condition can be achieved; It requires continued research on the destination's ecological, social-cultural, and economic systems [37, 38]. Therefore, sustainable tourism is not about ensuring the financial viability of tourism businesses and not just "greening" the tourism industry. Instead, sustainable tourism can be seen as a model of economic development [24, 39] to (1) Improve the quality of life of the host community, (2) provide a high-quality experience for visitors, (3) Maintain quality of the environment that both local people and tourists depend.

The relationship between tourism and community can be understood by a model consisting of four stakeholders: authorities, tourism business communities, local communities, and tourists [40, 41]. Local communities play a vital role in determining the success of a community-based tourism program. However, this success is challenging to realize without external support [42]. Tourism development should also focus on collaboration between local authorities and local communities through their participation in the tourism planning [42–44].

## Community based-tourism

Today, CBT has become a spearhead in support of rural development in both developing and developed countries. Generally speaking, CBT refers to the involvement of local communities in tourism activities by providing tourism services. It means they own, manage, and coordinate tourism activities while receiving direct economic benefits and protecting cultural and natural heritage [2, 45]. CBT is a type of tourism that needs to consider social, environmental, and cultural sustainability. It is operated and owned for and by the people to raise visitors' awareness and to teach them how to live locally and in the community [2, 42, 46].

Many studies confirm that CBT serves as a pathway to equal distribution of benefits to all individuals involved in tourism through a mutual agreement of self-determination and decision-making, co-subject responsible for development [41, 43, 47, 48]. It is found that actual consensus and natural, local control over development are practically not possible or desirable by particular communities. CBT is a form of sustainable tourism development [49]. In addition, CBT includes the high level of participation of local communities in the practice and development of projects or initiatives on development issues [2, 50].

Pearce, Moscardo, and Ross [51] argue that the lack of community capacity and tourism knowledge are obstacles to successful tourism development in emerging countries. Thus, enhancing the local people's capacity to participate in tourism activities is very crucial. The prominent role of CBT is to give communities, individuals, and organizations opportunities to form their capacities for CBT development as they are interested in and desire to be involved.

Community plays an essential role in sustainable tourism development. The studies argue that community compliance helps to reduce conflicts and misunderstandings between host

communities, local authorities, and tourists [2, 51]. In addition, community participation can reduce opposition to tourism development, minimize negative impacts and enhance the livelihoods of host communities [41, 51, 52]. Another way to help reduce conflict is through improved cooperation between residents and destinations that contribute to understanding local issues and promote knowledge sharing [41, 42]. Scholars argue that ignoring community input in tourism development can lead to conflicts between stakeholders, especially between communities and tourists, resulting in the decline of the destination tourism [2, 53].

Furthermore, the role of community participation is neglected in decision-making in CBT development due to some barriers from within and outside the community. Cole [53] pointed out that community participation is limited by institutional factors such as centralized decision-making, unwillingness to include host community members in their work, and the lack of knowledge about how to engage between host communities. Similarly, Mbaiwa [54] highlights the ignorance and the lack of skills and expertise stemming from insufficient training and education opportunities for local communities. In addition, other studies also show a loss of revenue, a lack of transparency in benefit sharing, and a lack of an appropriate policy framework for developing community initiatives to have a rewarding impact, including community participation in the tourism industry [2]. On the other hand, Cole [53] identified the lack of ownership, capital, skills, knowledge, and resources as barriers to the active functioning of local communities. However, Tosun [55] introduces three significant types of limitations covering all challenges, including limitations in activities (concentration of power, lack of coordination, lack of information), limited structure (lack of expertise, excellence, lack of trained human resources), and cultural constraints (limited capacity, indifference and low awareness of local people).

In Vietnam, sustainable CBT development is recently becoming a topic of great research interest. Most studies evaluate the current situation and solutions for CBT development in a particular locality or province. For example, Thuy [56], Hong [57], and Van [58] show that CBT needs to bring benefits to the community to ensure long-term development. Some studies have shown that coordination among stakeholders plays a vital role in sustainable CBT development, but most of this collaboration has received little attention [59, 60]. Meanwhile, Loi [61] argues that it is necessary to have specific policies and strategies to promote the development of community tourism. Without the State's policies and management, some households develop spontaneously, which may lead to a negative tourism image in the eyes of tourists. However, recent research is also focusing on product development by exploiting the strengths of specific local products to attract tourists and improve the lives of the community by selling souvenirs for tourists [59, 60]. At the same time, a factor that is currently being focused on is that most current models of community-based tourism cannot promote communication and marketing to attract tourists [61]. Thus, these studies are an essential foundation for identifying the common problem of sustainable CBT development. These are valuable studies to compare and contrast with the case of community tourism in Nhon Ly fishing villages of Binh Dinh province.

In sum, CBT is formed as a tool and solution to distribute equally benefits to all individuals engaged in tourism, adoption of a mutual agreement on decision-making, and the subjective community responsible for development. CBT includes a high level of community participation in the tourism project and development, the control of local communities over the development process, and the direct benefit for the local communities.

## 3. Study site and context

Located on the South Central Coast of Vietnam, Binh Dinh province has about 6,022 square kilometer and 1,486,465 people [62]. The area also owns a coastline of 134 km and is

considered an excellent potential for coastal tourism. In recent years, Binh Dinh tourism has seen rapid development, becoming one of the most important economic sectors contributing to the province's economic growth. According to General Statistics Office [62], the tourism industry of Binh Dinh province has dramatically contributed to the locality's socio-economic development. The total contribution to tourism accounts for 20%.

Binh Dinh has become a safe, friendly destination, attracting many domestic and international tourists to visit and stay on an extended vacation. In June 2020, Quy Nhon was ranked in the top 20 destinations in the world 2020, organized by *Hostelworld*. According to General Statistics Office, there has been a considerable growth in provincial tourism from 2017 to 2019. Table 1 shows the number of tourists and revenue from tourism in Binh Dinh in 2016–2019.

Nhon Ly, a coastal commune, is located in the eastern part of the Phuong Mai peninsula of Quy Nhon city, Binh Dinh province. It has an area of 12,13 square kilometer and a population of 7,709 people [62]. Nhon Ly commune is divided into four villages: Ly Chanh, Ly Hoa, Ly Luong, and Ly Hung. People in the commune still live in marine-related occupations such as fishing, seafood processing, and animal husbandry. This coastal commune still preserves many Champa cultural heritage. Besides, this locality also has much potential for developing coastal tourism and CBT. The landscape of the fishing village, traditional fishing, beautiful beaches, and Eo Gio have become attractive natural and human resources, attracting visitors to Nhon Ly (Fig 1).

Recently, many families in Nhon Ly have changed from fishing to participating in tourism services. Tourism development has helped a part of fishers get more jobs from tourism services. Nhon Ly is in the "involvement" stage of the tourism life cycle; thus, clearly identifying current tourism development problems will be essential to propose solutions to supplement the most suitable tourism development strategy for Binh Dinh province.

Binh Dinh province has implemented CBT programs in Nhon Ly to promote the cultural and ecological values for the development of CBT. In early April 2018, the Department of Tourism cooperated with the People's Committee of Quy Nhon City to organize a symposium "Preservation of culture and architecture of coastal villages for CBT development in Nhon Ly commune" to discuss developing the CBT on this site. On November 2, 2020, the Provincial People's Committee confirmed the deployment of the zoning plan of 1/2000 scale titled "Conservation and Development of sustainable tourism in the area of Nhon Ly fishing villages" to preserve and promote the traditional architectural, landscape, and cultural values of the fishing village and to build a model for sustainable tourism development.

**Table 1. Number of visitors and revenue from Binh Dinh tourism in the period 2016–2019.**

| No.tourist and revenue | Unit | Year | | | | |
|---|---|---|---|---|---|---|
| | | **2016** | **2017** | **2018** | **2019** | **2020** |
| **Total of tourist** | Person | 3.200.000 | 3.700.045 | 4.092.340 | 4.829.000 | 2.222.500 |
| *International* | " | *256.000* | *264.470* | *327.380* | *484.000* | *144.600* |
| *Domestic* | " | *2.944.000* | *3.435.575* | *3.764.960* | *4.345.000* | *2.077.900* |
| **Growth rate** | % | | 15,6 | 10,6 | 18 | -54 |
| *International* | " | | *3,3* | *23,8* | *47,8* | *-70* |
| *Domestic* | " | | *16,7* | *10* | *15,4* | *-52,2* |
| **Revenue** | USD (million) | 64,4 | 91,8 | 172 | 258 | 101 |

Source: Binh Dinh Department of Tourism (2021)

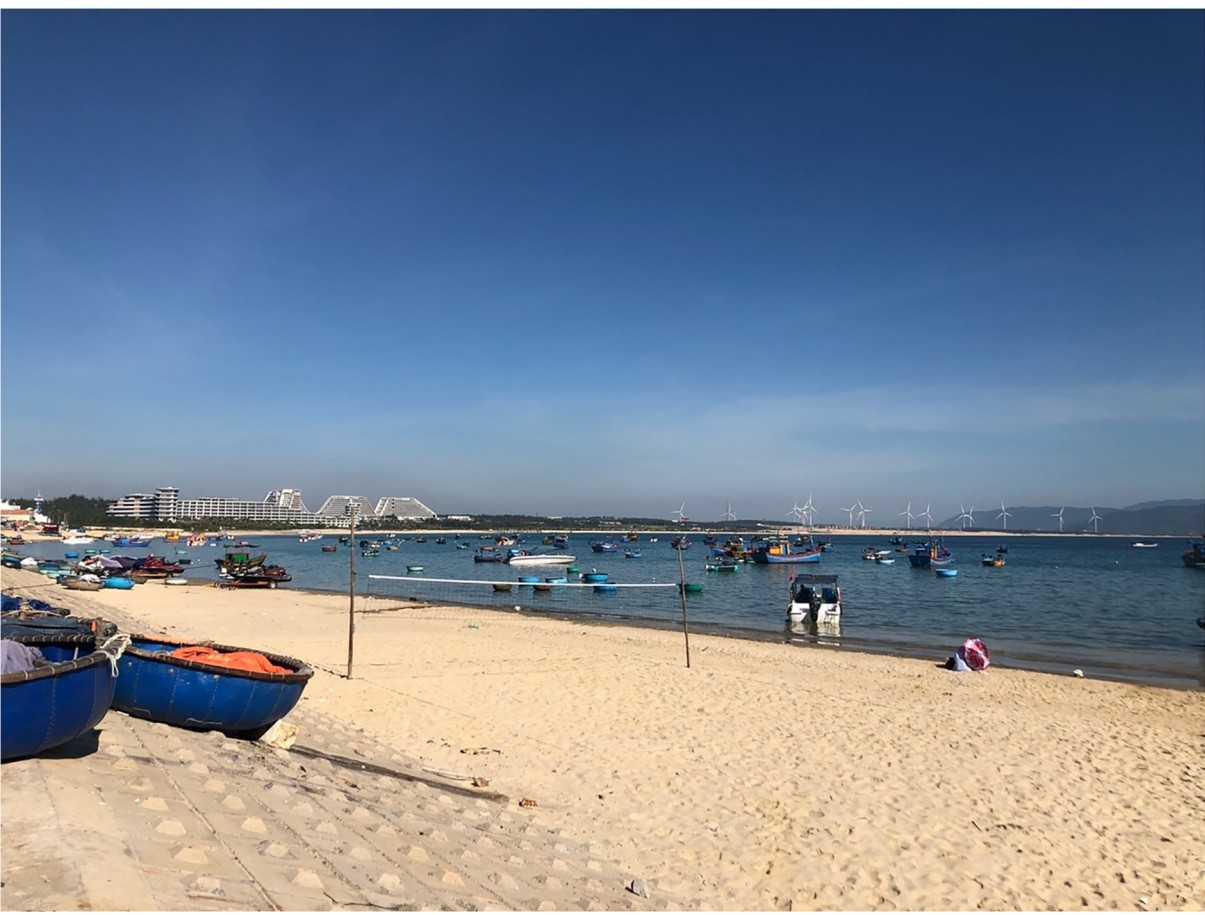

**Fig 1. A corner of Nhon Ly (photo courtesy: Author).**

Overall, thanks to tourism's economic development orientation, Nhon Ly, a poor coastal commune, has made substantial changes. This shift shows that Nhon Ly commune has a solid capacity to develop tourism, especially CBT.

## 4. Research methodology

This study employed several methods, including secondary data, Participatory Rural Appraisal (PRA), and interviews. Firstly, secondary data was collected based on reports, documents, policies, and resolutions related to Binh Dinh province's economic and tourism development. At the same time, primary data was collected based on PRA and semi-structured interviews through fieldwork.

Participatory Rural Appraisal is used to record the local experiences and their daily life while participating in activities with local participants or key informants [63]. For us, the PRA method is considered a fundamental method in research to assess the current situation of tourism development in Nhon Ly-Binh Dinh in recent years. In this study, we observed daily life activities and specialized professional activities in the 12 months of 2022 to have an insight into people's lives from the local perspective while maintaining an outsider's view as a researcher. During the research process, the author also wrote daily field notes about any information, conversation, interactions, or events considered essential information for the study [64].

Semi-structured interviews were used to collect data through interviews with various participants in Nhon Ly. Such information providers allowed the authors to understand their views on issues in the development of CBT [64, 65]. Semi-structured interviews were conducted with local communities, including four main groups: canoe-tour services, homestay services, grocery stores, and restaurants (Table 2). They were encouraged to talk freely about their perceptions and attitudes toward community-based tourism based on their experiences [65]. Each interview was spent 45–60 minutes at any convenient location at the participant's homestay, restaurant, boat, home, and public cafe. To avoid misrepresentation, we used two digital devices to record during the interview. In addition, supplementary notes were used during and after the discussions to ensure the data triangulation [64, 66].

Interviews were recorded on two digital devices, and field notes were taken to enable comparison and avoid misrepresentation. We used a thematic approach to the data analysis (Stake, 2010). Those materials (field notes and interviews) were then transcribed and coded to identify the themes [64, 65]. Finally, we highlighted and merged the sentences and critical terms to establish categories and descriptive statements, allowing us to identify data patterns, formulate meaning, and discover relationships among them.

Regarding the approval of the ethics committee of our study, so far in the case of Vietnam, to our best knowledge, there are no ethics committee practices in the majority of Vietnamese universities. This research is part of the author's dissertation, carried out at Nguyen Tat Thanh University, Vietnam where there is no ethics committee. However, our research proposal and the research result were reviewed and approved by the Evaluation Committee Members of Nguyen Tat Thanh University. Moreover, during the field trip, before conducting the interviews with the potential participants, the authors asked permission from the participants, if they agree to be interviewed, the author would begin the conversation, otherwise, the author would ask the others. The interviewees freely withdraw from the interviews anytime. This

**Table 2. Profile of respondents.**

| No | Code | Tourism-related job | Gender | Age |
|----|------|---------------------|--------|-----|
| 1 | LOC001 | canoe-tour services | Male | 61 |
| 2 | LOC002 | canoe-tour services, seafood selling | Female | 42 |
| 3 | LOC003 | canoe-tour services | Male | 39 |
| 4 | LOC004 | homestay services, canoe-tour services | Female | 33 |
| 5 | LOC005 | restaurant, homestay | Female | 54 |
| 6 | LOC006 | grocery stores | Male | 40 |
| 7 | LOC007 | grocery stores | Male | 60 |
| 8 | LOC008 | canoe-tour services | Male | 32 |
| 9 | LOC009 | homestay services | Female | 45 |
| 10 | LOC010 | homestay services | Male | 57 |
| 11 | LOC011 | grocery stores | Male | 43 |
| 12 | LOC012 | grocery stores | Male | 49 |
| 13 | LOC013 | restaurant | Female | 35 |
| 14 | LOC014 | canoe-tour services | Male | 32 |
| 15 | LOC015 | homestay services | Female | 45 |
| 16 | LOC016 | canoe-tour services | Male | 49 |
| 17 | LOC017 | canoe-tour services | Male | 45 |
| 18 | LOC018 | canoe-tour services | Male | 57 |
| 19 | LOC019 | canoe-tour services | Male | 43 |

meant, we obtained verbal consent from all interviewees in our study. To protect the interviewees' privacy, we used the respondents' codes to reference their statements in the findings.

## 5. Findings

The results are presented in four parts: the profile of interviewees, the impact of tourism on the local people's lives, the level and type of community participation in tourism, and the critical challenges of the CBT development in Nhon Ly.

### Profile of interviewees

Of the 19 interviewees, 13 were men (68.4%). Respondents' mean age was 45.3 years, and their jobs were related to canoe-tour services (9), homestay services (4), grocery stores (4), and restaurants (2).

### Impact of tourism on the local people's lives

The results showed that, from 2016 to 2020, a strong tourism boom brought positive changes to the local economy. The local community's life has improved; many fishing households have changed from fishing to tourism services. Previously, most of the villagers in Nhon Ly worked as fishers, their lives were still tricky, but the appearance of tourism has made their lives more prosperous and dynamic. One fisherman shared:

> In the past, my family relied mainly on seafaring for a living. However, since tourism has developed in our area, I became interested in it and started operating fishing tours. This allowed me to earn money and have someone to share my seafaring experiences with
>
> (LOC001).

Similarly, a resident selling seafood to tourists also said:

> Since many tourists have started coming here, my family's income has changed positively. We sell seafood purchased from local fishers to tourists on a large scale. The seafood is always fresh since we buy it directly from the fishers, which the customers love. Moreover, the selling price is perfect compared to market sales.
>
> (LOC002).

Tourism development has also boosted families' business in this sector for the better. Many households have set up their tourism businesses and organized experiential tourism activities to increase their income. Many local people believed that Nhon Ly had great natural and cultural resources for tourism development. A travel service business owner who is one of the green tour promoters mentioned:

> Nhon Ly is an attractive destination that has great potential for the exploitation of its natural resources, combined with the cultural value of the fishing village. As a resident of Nhon Ly, I am eager for the development of my homeland and the preservation of the cultural values of the fishing village. Therefore, my friends and I decided to design a tour that prioritizes sustainable factors and environmental protection. Since then, tours associated with environmental protection have been introduced
>
> (LOC003).

It is clear that local people, including local business households, also pay attention to protecting their homeland's natural resources and environment in the long term. A homestay owner pointed out:

When starting our tourism service in Nhon Ly commune, we also placed great emphasis on connecting with the local community for mutual development. Initially, we succeeded in attracting residents to participate in a joint campaign aimed at protecting the environment and the coastal coral reefs. The sense of community has had a positive impact on many tourists, who have shown a keen interest in our environmentally-friendly tours

(LOC004).

Thus, the members of the local community here, especially tourism service businesses, have also focused on protecting nature, raising the general awareness of tourists and local people. It also connects members participating in local tourism activities with each other. Some respondents talked:

At first, my green tours focused on promoting a civilized code of conduct in tourism and raising awareness of environmental protection at the destinations among tourists. Later on, we started to actively involve our visitors in local environmental protection activities such as cleaning up the area, collecting plastic bottles, and exchanging gifts. They also had the opportunity to gather shells to make souvenir products by hand. Many tourists and residents have attended the workshops organized by several tourist service establishments

(LOC003).

In 2020, when the community group for the management of fisheries resources in Nhon Ly became officially operational, the green tours had more specific activities to protect aquatic resources, particularly the coral reefs on the seabed. We are aware that tourism activities can have an impact on the marine ecosystem. Therefore, when organizing scuba diving tours, we pay close attention to compliance with regulations to ensure that visitors have an enjoyable experience while also making sure that it does not adversely affect the marine ecosystem

(LOC004).

The results also show that many families have invested in developing homestay services to bring more experiences to tourists (Fig 2). This business not only generates income for family members but also promotes and stimulates local people's general entrepreneurial spirit. As a result, homestays have also emerged, creating many job opportunities and improving the livelihoods of the community. Examples of these homestays include Homestay Thien Phu, Moc Homestay, La Beach House 1 and 2, Eden Home, Lan Anh Homestay, and Havas Homestay. A local homestay operator—who used to be a guest traveling to Nhon Ly but loves the life here, sharing about participating in local tourism activities:

I had a special connection with Nhon Ly and adored the peaceful beauty of the fishing village. That's why I established a chain of green-friendly homestays that were in harmony with the architecture of the fishing village. The tranquil charm of Nhon Ly's fishing village was a valuable resource, and it was precisely why I chose to stay there. When building homestays, my core objective was to create accommodation that blended in with the architecture and landscape of this beautiful fishing village

(LOC004).

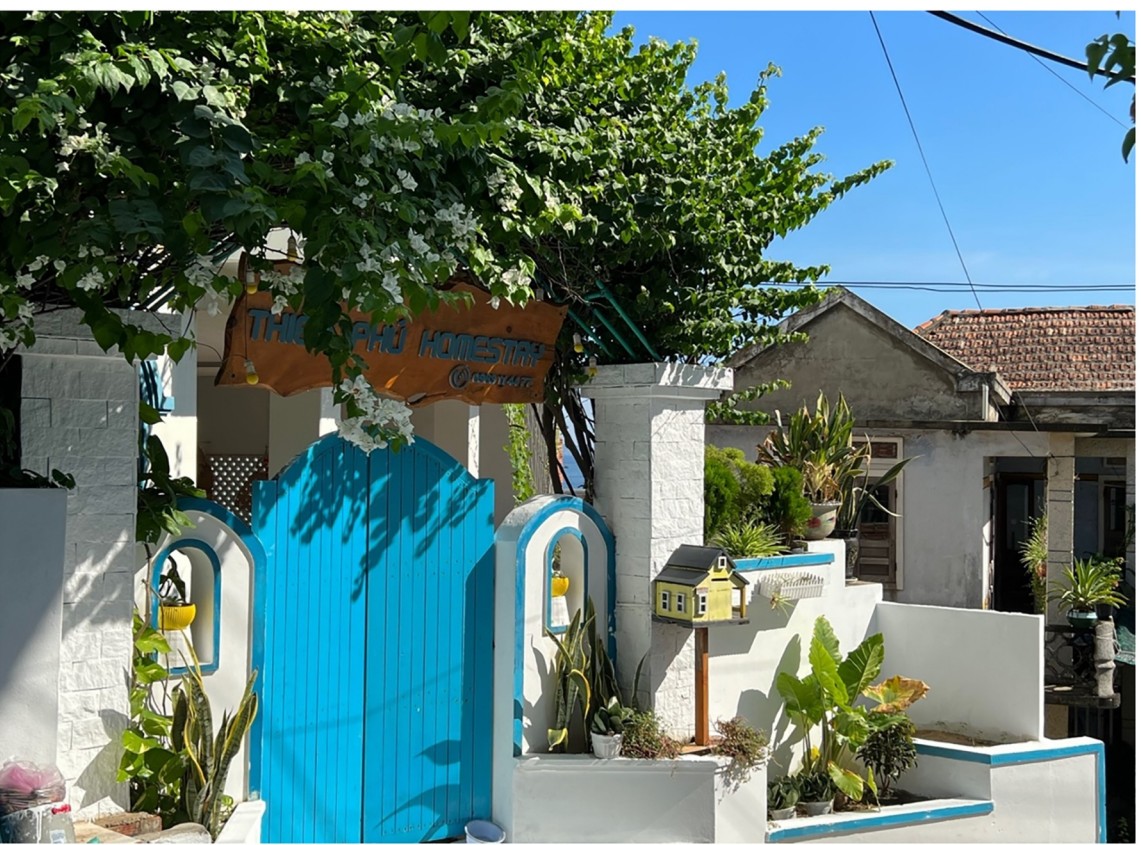

**Fig 2. A homestay in Nhon Ly (photo courtesy: Author).**

Some households' bravely pioneering the booming tourism business has contributed to creating a positive change for many other people in Nhon Ly to invest, or associate to support in more canoes and restaurants, homestays to serve guests. A local restaurant businesswoman said:

Apart from focusing on the restaurant business in the Eo Gio area, my family also invested in 2 other canoes. In addition, I also cooperated with the homestay for guests to stay

(LOC005).

A homestay owner shared:

Because my house was in a nice view—near the Eo Gio area, I have invested more than 1 billion VND (about 43,000 USD) to renovate and build a house from 1 floor to 3 floors with single rooms, a gym to serve guest

(LOC004).

Based on the author's observation, accommodation owners in Nhon Ly are much-paid attention to improving service quality and satisfying visitors. Many tourism businesses are engaged in CBT, and members of many households always try to keep the living room, garden,

and area around the house clean to attract visitors. In addition, in Nhon Ly, fishers' households who work as amateur tourism services are not rare but also attract tourists' excitement for the sincerity of the anglers. The study also shows that fishing boats used by anglers for fishing have now been converted into ships carrying tourists for sightseeing and fishing. Besides, the number of tourist canoes in the commune is increasing significantly. These services have prevented fishers from going through turbulent sea trips to make a living, but now they are pretty gentle for tourists to fish and squid fishing. With intangible resources regarding marine life, the sincerity, gentleness, and the wild beauty of the coastal area, households in Nhon Ly are changing their lives based on tourism services.

The results also showed that, since the beginning of 2015, because many tourists, entertainment, and resort areas appeared massively in Phuong Mai Peninsula, tourists started to flock to Nhon Ly for many beautiful and peaceful scenes. As realizing the opportunity for development, many households open a tour to take tourists to visit. According to some families:

> Initially, we converted a fishing boat into a tourist boat to take guests to Eo Gio, Ky Co, and Hon Kho to enjoy fresh local seafood. However, the business was a bit unstable because not many people specialized in tourism, and we were afraid of customer complaints. However, guests appreciated our sincerity and continued to return here many times. They also recommended more friends to visit here, making our tours more popular and attracting many tourists to our hometown
>
> (LOC007).

> With the tourism boom in Nhon Ly, tourists from other regions started visiting the place frequently. As a result, people came together to borrow capital to convert boats, purchase canoes, and build more spacious restaurants to cater to tourists. Initially, many people were hesitant to invest in tourism services, and only a few family members took the plunge. However, as more people succeeded and expanded their operations, they began to hire more local workers and increase their scale of operations
>
> (LOC004).

In addition to being self-employed in tourism services, many young people actively participate in some activities of the Department of Tourism and Tourism Association of Binh Dinh province to promote tourism in Hanoi, Quang Ninh. One local tourism business owner shared:

> People like me were interested in learning and promoting the tourism image of our locality. Therefore, I enjoyed participating in government authorities' activities to promote tourism to other provinces. These trips outside the area helped me learn many valuable things about doing business from successful cases in different localities. By doing so, I was able to reflect on what I had done and what I had yet to achieve, so I could overcome challenges more effectively in the future
>
> (LOC006).

In general, tourism in Nhon Ly has developed and contributed significantly to the economic changes in the community; a part of fishers have changed from traditional jobs for generations to the provision of tourism services. In addition, the tourism business households have also consciously protected marine resources through many activities of preserving sea sanitation and opening green tours. This transformation not only improves fishers' livelihoods

but also opens opportunities for more households to have more confidence to participate in local CBT services.

## Level and type of participation in tourism

Community participation in tourism plays a vital role in the success or failure of a sustainable CBT model [51, 67]. The research shows that the local community in Nhon Ly has gradually participated in providing tourism services such as homestay business, catering, sightseeing, sales, and souvenirs. They enjoy a part of the benefits in the value chain of tourism. The community is active in tourism activities and plays a role as both "tourism resource" and "tourism resource developers" in their homeland. The research results show the main activities in tourism service provision that people participate in (Table 3).

Thus, due to the lack of specific CBT policies and plans, the community in Nhon Ly is not the organizer; therefore, the central values of CBT are not yet fully promoted. Most of the people doing CBT are spontaneous. Through our fieldwork, community participation of Nhon Ly is only at the level of service provision, not participating in the decision-making process and building implementation plans in CBT. Therefore, the community in Nhon Ly has tourism activities but only stops at the second level, "Tourism with community participation".

With favorable traffic conditions, the number of tourists flocking to Nhon Ly tourism increased in size, from a poor fishing village now is changing positively. Most anglers used to depend on seafaring and fishery production, but now they practice tourism. Our fieldwork exploring the type of tourism service participation by local communities shows various sectors and forms of community participation in tourism (Table 4).

**Table 3. Descriptions of level and type of participation in tourism by people.**

| Level of participation | Type of participation |
|---|---|
| 1 | Primarily involved in providing local products |
| 2 | A large number of people are involved in accommodation, catering, and transportation services |
| 3 | Very few people are allowed to participate in contributing ideas to the authorities at all levels |
| 4 | Very few residents participate as guides, or if any, are just "companions" with tourists rather than a professional guide |
| 5 | Almost no people participate in the organization and management of tourism in Nhon Ly. |

(Source: compiled from authors' survey, 2021)

**Table 4. Description of sectors and types of participation in tourism by people.**

| Sector | Types of participation |
|---|---|
| Transport service | Join in tour guide activities |
| Commodity business | Sales and production of handicrafts |
| Catering service | Agricultural and aquatic production for tourists |
| Accommodation service | Homestay |

(Source: compiled from authors' survey, 2021)

This research also showed that: (1). Most households produce and supply agricultural and aquatic products not only for tourists but also for restaurants and floating restaurants; (2). Some families engage in food and beverage business in a range from small restaurants to catering services on the floating restaurant; (3). Many households in the sea transportation business focus on renting boats or transporting canoes to serve tourists visiting the bay, diving and watching coral, and experiencing squid fishing; (4). Home-based business activities and community participation are mainly in Ly Luong village (i.e., one of the four villages in Nhon Ly); (5). Other activities include travel business, tour guide activities, or handicraft production with few participants.

The abovementioned findings showed an imbalance in the provision of tourism services and inequality regarding tourism development between villages, and the local workforce was not trained enough to meet the high demand of tourists.

## Challenges of the CBT development in Nhon Ly community

The findings showed that the majority of tourism services provided by local people in Nhon Ly were still spontaneous. Many families worked in tourism, which led to inadequacies, such as causing difficulties in administrative management and negative impacts on society and the local environment. By approaching local community perspectives, the study results showed:

Firstly, in terms of tourism policy: policies related to the development of CBT were not issued and implemented in a specific way. Although local authorities proposed many programs and projects on CBT development, their implementation in local communities was quite limited. Most of the policies were on paper and had not been implemented yet. Hence, the community did not fully understand their role, the benefits of CBT development, capacity-building policies, and supporting funds. Therefore, the development of CBT was almost exclusively taking place below the spontaneous community level.

Secondly, with regard to administrative management, there was no consensus on establishing a destination management board, managing social security, residence, and direction management in Nhon Ly. The development of tourism has not concretized the support for the participants. In particular, the members of the management board, such as Nhon Ly Tourism —Fisheries Trade Service Cooperative, mainly belonged to the local government, not the local community. Additionally, the agency had its headquarters outside of Nhon Ly commune. Therefore, the community faced both geographical distances and reflected problems in tourism. Furthermore, the distribution of benefits between the parties and contributions to the environmental protection fund remained unclear, maintaining traditional activities. As a result, the management board's role has not been entirely fulfilled in promoting the conditions for CBT in Nhon Ly.

Thirdly, concerning the environmental aspect, some local businesses have placed a strong emphasis on preserving natural resources and promoting ecological sanitation through various green tourism activities and awareness campaigns for the community and tourists. However, the rapid growth of tourism and lack of proper control has resulted in a significant decline in fisheries resources in the bay area. Additionally, the large number of operating canoes has contributed to water and air pollution. As a result, long-term utilization of natural resources should be prioritized when considering the environmental factors in Nhon Ly.

Fourthly, with regards to the socio-cultural aspect, the rapid tourism development in Nhon Ly has led to competition among local businesses, particularly in the restaurant sector, resulting in a decline in service quality and price disparities. Additionally, some residents have reported problems with rowdy and drunk young people in the area, causing disturbances and driving recklessly in residential areas, which raises concerns for the safety of both locals and

tourists. The increase in land prices has also resulted in conflicts over land rights and disrupted the traditional social harmony of the coastal fishing village. However, the tourism industry of Binh Dinh province is promoting a green, clean, friendly, and safe image for the local tourism sector. Failure to address these socio-cultural issues may not only hinder the tourism development of Nhon Ly but also damage the province's reputation as a safe and welcoming destination.

Overall, despite the limitations of spontaneous tourism development, the fishing village community of Nhon Ly in Binh Dinh province had certain advantages for development. Specifically, the community experienced an improvement in their quality of life through the new revenue source generated by tourism. The increase in cultural awareness among locals, as they enhance their professional expertise to meet visitors' needs, was another advantage. Additionally, relevant departments of Binh Dinh province have initiated policies in planning and new urban development to meet the development needs of the local profession, including tourism.

## 6. Discussion

In general, this study showed that tourism positively impacts the economy, socio-culture, and environment of the Nhon Ly community. Similar to previous studies [41, 48], this study demonstrated that tourism has contributed significantly to improving the lives of local people and promoting local engagement in tourism activities. However, several challenges have happened during the development process in Nhon Ly fishing villages. This study found that the causes of these shortcomings can be related to: (1). Awareness and capacity of community participation in tourism are limited; (2) Policies to encourage community participation in tourism have not been effectively implemented; (3) Qualified human resources for CBT are lacked; (4) The community has not equally benefited from tourism.

This research also showed that the organization of propaganda, mobilization, and training of locals to participate in tourism activities was still complex. Most local communities believed that investing in infrastructure, technical materials, and preserving and preserving traditional cultural values is the work of the government and tourism businesses. People were not consciously self-aware of participating in local tourism development. In particular, they have not yet recognized their role, position, and importance in exploiting and conserving local tourism resources. There was no consistent development link between the parties involved in tourism activities. The findings of this study are consistent with Lenao's [68] study when it discusses the challenges of tourism development in Lekhubu, which include poor accessibility, limited income generation, over-reliance on funding from outside, inadequate marketing as well as unsatisfactory involvement, and limited capacity among board committee members.

Many previous researchers have discussed the role of the community in the decision-making process in tourism planning and the importance of CBT projects' implements, especially the establishment of a local management board to support local communities in solving tourism issues [53, 69–73]. However, our research demonstrates that the local government has not yet set up a management board in which people are directly involved in drafting the plan and defining the project's goals and benefits. Although the province has a plan for the Nhon Ly area to develop CBT, it has not yet implemented it extensively and effectively for households doing tourism business. Moreover, the stated criteria have not been applied yet, so tourism is ineffective. Sometimes people still have to go back to fishing to stabilize their lives. In particular, when conducting tourism business, local households also encounter cumbersome legal administrative procedures, high taxes, and fees, causing significant obstacles for them. Studies have shown that government should play a fundamental and specific role by formulating, implementing, and monitoring policies and laws to foster a shift in the local direction [69, 71–

73]. More controlled tourism can enhance coastal self-reliance, redistribution and equality, and the well-being of communities and individuals [20, 36, 74].

Furthermore, the government and other organizations should, through funding pro-CBT initiatives, enable local disadvantaged communities to participate in the tourism sector [75–77]. Therefore, it is necessary to have appropriate mechanisms and policies to encourage and ensure CBT activities in Nhon Ly communities develop in the right direction and maximize efficiency. In addition, our research also shows that the overlapping of management groups has constrained and confused the ability to practice, control and supervise locals among government management boards. Thus, the community also becomes passive when asking for advice from such agencies. In particular, the overlap among the Ministry of Culture, Sports and Tourism; Department of Tourism of Binh Dinh Province; People's Committee of Quy Nhon city—People's Committee of Nhon Ly commune; Community Tourism Management Board, and Residential Community. Thus, we argue that planning and building a management mechanism must ensure the proper management system, the linkage between the entity and the appropriate management capacity, and the management model must be decentralized to avoid overlapping these agencies.

During our research, we found that the lack of professional skills has significantly limited the ability of the locals to serve visitors [53, 72]. Although understanding geography, culture, and environment in the locality, the locals do not have good social knowledge, understand tourist culture, and limited language proficiency, causing significant obstacles to serving tourists. In addition, CBT in this area has only been formed in the past few years, so the locals have not well enhanced their capacity to organize tourism activities and member cohesion. Furthermore, people have not received support from the authorities in building models for CBT, and loan support for tourism purposes is also limited. Therefore, locals still have to do tourism business in poor conditions, and homestays sometimes lack equipment, lighting systems, clean water sources, and sanitation around the environment that are not guaranteed for tourists. With active support, these CBTs will promote the great value and benefit all parties [41, 59, 78].

The only way to overcome the above obstacles is when planning for CBT development; it is necessary to consider community participation's operational, structural and cultural limits [72, 76]. Participation at the local level is essential to achieving the global goal of sustainable development [79]. However, such participation often involves a power shift from local government to local actors. Moreover, real consensus and absolute local control are not always possible. Planners need to understand the complex networks of power shift relationships and how different stakeholders envision CBT [78]. Primary education and training are required for the target communities to accompany tourism development. Local communities must develop strategies for receiving and interacting with tourists and expressing themselves and their tangible culture [80]. This solution involves finding the right balance between economic benefits and cultural integrity.

Finally, our research demonstrates that tourism has not brought jobs and stable and synchronous income because many fishers are used to sticking with the sea profession; tourism has only developed in recent years. Therefore, capital sources for tourism investment are mostly the people's self-owned capital; there is no policy to support prosperity from other sources (e.g., governments, banks, businesses). Therefore, some households are participating in tourism as hired labor and seasonal. Families participating in certain activities in the tourism business are not sure about the profits the company gets from tourists and are only paid for completing the assigned tasks [73]. In addition, some households associate with travel agencies; the company only pays a small commission for the connection points, and the rest has other costs that people have to spend on their own without receiving the sharing. As a

result, tourism income has not yet reached all households. Therefore, equitable benefit-sharing between the participating parties is critically needed [43, 74, 81].

In summary, the CBT is gradually taking shape in coastal localities and is giving visitors some exciting experiences. Community-based tourism is a suitable direction so that coastal localities can find economic benefits and contribute to preserving traditional cultural values. Under fishers' uncertain natural living conditions, creating new forms of sustainable livelihood has become a top concern. Therefore, CBT development can be considered an appropriate way. Through this tourism development, the coastal communities will benefit in many ways, thereby improving the quality of life with opportunities involved in CBT.

## 7. Conclusion

This study explores the impact of tourism on the lives of the local community, the level of people's participation in tourism, and the challenges of tourism development in the coastal commune of Nhon Ly, Vietnam. This study has shown that tourism positively impacts various aspects of the local community. In addition, communities are strongly aware of the importance of marine resources for ensuring their livelihoods in tourism development. Therefore, most community members actively participate in conservation activities to protect the marine environment. This study contributes to understanding a different perspective of coastal fishers in Vietnam. While in many places, the environmental impacts on beaches and islands are alarming [9, 82] in Nhon Ly destination, the community is aware of the importance of marine resources for the security of their livelihoods in tourism development. Therefore, most community members have a positive attitude towards conserving the marine environment. This study affirms the importance of a sense of community along with their self-governing activities. The community is considered an essential factor in sustainable tourism development in terms of environmental protection and in ensuring the community's sustainable livelihoods.

Another contribution of this study is to bring a more positive perspective on the formation of tourism business groups on the local community's land. In the past, the State and tourism business groups implemented several projects, and the locals protested extremely strongly because they thought their land area was shrinking. Their living space was also negatively affected. Previous studies have shown positive and negative aspects of tourism investment projects [83–86]. In some cases, the locals see positive tourism changes, but they feel ambivalent about tourism's impact [87, 88]. However, our research findings show that after more than ten years of implementing a tourism project in Nhon Ly, tourism has boomed and created positive changes in the economy and socio-culture of the community and become a significant driving force to stimulate the development of the locality. If the tourism development projects of the State and investment groups cause damage to the economy, culture, society, and environment of the community, indeed tourism development activities here will not be sustainable. Thus, the community's attitude will become increasingly fierce in opposition to such development. However, in this study, we affirm that the formation of resorts by tourism business groups and the development of CBT can develop in parallel to create various products and options for tourists in a destination.

This research involved small samples of host communities in the Nhon Ly community, which may not be truly indicative of the perspectives of all communities in emerging tourist countries. Despite carefully selected samples based on considerations, they might not represent the complete image of local communities. In addition, it is vital to refer that the limitations of our research are due to the Covid-19 constraints, which have prevented us from taking more samples and conducting other approaches such as mix-methods or quantitative approaches. Thus, it is noted that, as this is a convenience sample, it may not be generalized to the more

significant population in different areas and peoples. Another limitation of this study is that it only approaches local perspectives. Therefore, the perspective may not reflect the different dimensions of community tourism and the participation of stakeholders such as the State, tourism businesses, and especially investment groups. Future studies may consider additional aspects of the barriers to stakeholder coordination or co-creation of tourism destination value.

## Author Contributions

**Conceptualization:** Tuyen Dai Quang, Hoang Van Nguyen.

**Data curation:** Tuyen Dai Quang, Quynh Xuan Thi Nguyen, Hoang Van Nguyen, Vang Quang Dang, Nhan Thong Tang.

**Formal analysis:** Tuyen Dai Quang, Quynh Xuan Thi Nguyen, Hoang Van Nguyen, Vang Quang Dang, Nhan Thong Tang.

**Funding acquisition:** Tuyen Dai Quang.

**Investigation:** Tuyen Dai Quang, Quynh Xuan Thi Nguyen, Vang Quang Dang.

**Methodology:** Tuyen Dai Quang, Quynh Xuan Thi Nguyen.

**Resources:** Quynh Xuan Thi Nguyen.

**Writing – original draft:** Tuyen Dai Quang, Hoang Van Nguyen.

**Writing – review & editing:** Hoang Van Nguyen.

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
