## [Decision Letter · Decision Letter 0]

19 Apr 2023

PONE-D-22-32881Toward sustainable community-based tourism development: Perspectives from local people in Nhon Ly coastal community, Binh Dinh province, VietnamPLOS ONE

Dear Dr. Hoang Van Nguyen

Thank you for submitting your manuscript to PLOS ONE. After careful consideration, we feel that it has merit but does not fully meet PLOS ONE’s publication criteria as it currently stands. Therefore, we invite you to submit a revised version of the manuscript that addresses the points raised during the review process.

ACADEMIC EDITOR: Please work according the the suggestions/ recommendations of Reviewer 2 to resubmit your article for consideration. There are some grammar mistakes and errors highlighted by the second reviewer. Please ensure that your decision is justified on PLOS ONE’s publication criteria and not, for example, on novelty or perceived impact.

We look forward to receiving your revised manuscript.

Kind regards,

Sakul Kundra, PhD in History

Academic Editor

PLOS ONE

3. We note that [Figure 1] in your submission contain [map/satellite] images which may be copyrighted. All PLOS content is published under the Creative Commons Attribution License (CC BY 4.0), which means that the manuscript, images, and Supporting Information files will be freely available online, and any third party is permitted to access, download, copy, distribute, and use these materials in any way, even commercially, with proper attribution. For these reasons, we cannot publish previously copyrighted maps or satellite images created using proprietary data, such as Google software (Google Maps, Street View, and Earth). For more information, see our copyright guidelines: http://journals.plos.org/plosone/s/licenses-and-copyright.

Natural Earth (public domain): http://www.naturalearthdata.com/.

4. "We note that Figures 3 and 4 includes an image of a [patient / participant / in the study].

Reviewers' comments:

Reviewer's Responses to Questions

**Comments to the Author**

1. Is the manuscript technically sound, and do the data support the conclusions?

Reviewer #1: Yes

Reviewer #2: Yes

2. Has the statistical analysis been performed appropriately and rigorously? 

Reviewer #1: Yes

Reviewer #2: Yes

3. Have the authors made all data underlying the findings in their manuscript fully available?

Reviewer #1: Yes

Reviewer #2: Yes

4. Is the manuscript presented in an intelligible fashion and written in standard English?

Reviewer #1: Yes

Reviewer #2: Yes

5. Review Comments to the Author

Reviewer #1: Understading the needs of the community in terms of development goals is the need of the hour.

Many times the research does not solve the problems of the grassroot level.

The research carreid out by the researchers solves a greater purpose in understanding the difficulty and the possible opportunity in uplifting lifes of people who are depened solely on the sea for thier lifestyle.

Any development has its pros and cons, the researchers have carefully studied the impacts of booming tourism as it brings in revenue as well as the challenges due to the increase of tourism.

A qualitative method best suited for this type of research brings out the essence of planning at the base level.

The research paper provides a great opportunity for the local authoritites and government to plan for the future tourism activities as it concentrates and provides possible outcomes by speaking to the local people with quality metrics rather than a quantity metrics.

Reviewer #2: The article cannot be published in its present form. My suggestions are as follows , after correction can be published:

a) There are grammatical mistakes and also problems in tense. A few I am mentioning: in 138 number line believe has been used should be reveal

b) Line 169 UNWTO full form should be used

c) 176 instead of currently presently should be used.

d) In 270 Km2 is not correct full should be used

e) In many places km has been used but in full form and must be used

f) In 315 line it should be research methodology

g) Also sampling may be used in this area

h) In 319, Primary Data conducted should be collected

i) In Table -2 it should be respondents

j) Table-3 and Table -4 are not correct it should be mentioned as Chart -3, chart-4

k) In the article some, many words have been used which should be avoided.

l) In the FINDINNS Present and Past tenses have been mixed. All MAINLY Should be in the PAST tense. FINDINGS SHOULD BE CORRECTED

6. PLOS authors have the option to publish the peer review history of their article (what does this mean?). If published, this will include your full peer review and any attached files.

Reviewer #1: **Yes: **N. Sriganeshvarun

Reviewer #2: **Yes: **Dr. Shankar Chatterjee

Former Professor & Head (CPME),NIRD & PR (Govt. of India), Hyderabad, India

While revising your submission, please upload your figure files to the Preflight Analysis and Conversion Engine (PACE) digital diagnostic tool, https://pacev2.apexcovantage.com/. PACE helps ensure that figures meet PLOS requirements. To use PACE, you must first register as a user. Registration is free. Then, login and navigate to the UPLOAD tab, where you will find detailed instructions on how to use the tool. If you encounter any issues or have any questions when using PACE, please email PLOS at figures@plos.org. Please note that Supporting Information files do not need this step.<quillbot-extension-portal></quillbot-extension-portal>

---

## [Author Response · Author response to Decision Letter 0]

25 May 2023

Dear Prof. Sakul Kundra, 

PhD in History

Academic Editor

PLOS ONE

Thank you for the opportunity to resubmit the manuscript. We have carefully reviewed the comments, suggestions, and questions from the reviewer. First and foremost, we would like to express our deep gratitude to you and the reviewers for providing constructive feedback and encouraging the publication of our manuscript. We have taken all of the concerns, questions, and corrections from the reviewers into full consideration to improve the manuscript. Please find our detailed responses to the reviewers’ comments below. We have also submitted the revised manuscript with track changes attached to this letter. Wishing you all the best!

Sincerely,

On behalf of the authors

Dr. Nguyen Van Hoang

We have checked the PLOS ONE style templates and followed the instruction accordingly.

We have included the ethics statement in the Methods section of our manuscript only.

3. We note that [Figure 1] in your submission contain [map/satellite] images which may be copyrighted. All PLOS content is published under the Creative Commons Attribution License (CC BY 4.0), which means that the manuscript, images, and Supporting Information files will be freely available online, and any third party is permitted to access, download, copy, distribute, and use these materials in any way, even commercially, with proper attribution. For these reasons, we cannot publish previously copyrighted maps or satellite images created using proprietary data, such as Google software (Google Maps, Street View, and Earth). For more information, see our copyright guidelines: http://journals.plos.org/plosone/s/licenses-and-copyright.

We have removed Figure 1 and replaced it with another figure taken by the author.

4. We note that Figures 3 and 4 includes an image of a [patient / participant / in the study].

We have removed Figure 3 and 4 from our manuscript

We have reviewed our reference list according to the guideline.

Reviewers’ comments

Reviewer 2’s comments Authors’ responses

a) There are grammatical mistakes and also problems in tense. A few I am mentioning: in 138 number line believe has been used should be reveal 

Thank you very much for your comments. We have corrected our manuscript based on your suggestions. Please see details in line 143 on page 7.

b) Line 169 UNWTO full form should be used 

Thank you very much for your comments. We have corrected our manuscript based on your suggestions. Please see details in line 176 on page 8.

c) 176 instead of currently presently should be used. 

Thank you very much for your comments. We have corrected our manuscript based on your suggestions. Please see details in line 186 on page 9.

d) In 270 Km2 is not correct full should be used 

Thank you very much for your feedback. We have made the necessary corrections to our manuscript based on your suggestions. Throughout the paper, we have written “square kilometers” instead of “Km2”

e) In many places km has been used but in full form and must be used 

Thank you very much for your feedback. We have made the necessary corrections to our manuscript based on your suggestions. Throughout the paper, we have written “kilometers” instead of “Km”

f) In 315 line it should be research methodology 

Thank you very much for your comments. We have corrected our manuscript based on your suggestions. Please see details in line 335 on page 15.

g) Also sampling may be used in this area 

Thank you for your comments. We appreciate your feedback but would like to clarify that we may not have fully understood this particular point. Our data was collected primarily through semi-structured interviews, and therefore we did not include the sampling method typically required for questionnaire surveys.

h) In 319, Primary Data conducted should be collected 

Thank you very much for your comments. We have corrected our manuscript based on your suggestions. Please see details in line 339 on page 15.

i) In Table -2 it should be respondents 

Thank you for your comment. We have corrected the table 2 based on your suggestions.

j) Table-3 and Table -4 are not correct it should be mentioned as Chart -3, chart-4 

Thank you for your feedback. We appreciate your suggestions, but we have decided to use Tables 3 and 4 as they provide supportive information in a tabular format.

k) In the article some, many words have been used which should be avoided 

Thank you very much for your comments. We have corrected our manuscript based on your suggestions.

l) In the FINDINNS Present and Past tenses have been mixed. All MAINLY Should be in the PAST tense. FINDINGS SHOULD BE CORRECTED 

Thank you for your comments. We are pleased to inform you that we have incorporated your suggestions into our manuscript, particularly in the Findings section.

---

## [Decision Letter · Decision Letter 1]

7 Jun 2023

Toward sustainable community-based tourism development: Perspectives from local people in Nhon Ly coastal community, Binh Dinh province, Vietnam

PONE-D-22-32881R1

Dear Dr. Nguyen,

We’re pleased to inform you that your manuscript has been judged scientifically suitable for publication and will be formally accepted for publication once it meets all outstanding technical requirements.

Kind regards,

Sakul Kundra, PhD in History

Academic Editor

PLOS ONE

Additional Editor Comments (optional):

Both reviewers have accepted the paper.

Reviewers' comments:

Reviewer's Responses to Questions

**Comments to the Author**

1. If the authors have adequately addressed your comments raised in a previous round of review and you feel that this manuscript is now acceptable for publication, you may indicate that here to bypass the “Comments to the Author” section, enter your conflict of interest statement in the “Confidential to Editor” section, and submit your "Accept" recommendation.

Reviewer #1: All comments have been addressed

Reviewer #2: All comments have been addressed

2. Is the manuscript technically sound, and do the data support the conclusions?

Reviewer #1: Yes

Reviewer #2: Yes

3. Has the statistical analysis been performed appropriately and rigorously? 

Reviewer #1: Yes

Reviewer #2: Yes

4. Have the authors made all data underlying the findings in their manuscript fully available?

Reviewer #1: Yes

Reviewer #2: Yes

5. Is the manuscript presented in an intelligible fashion and written in standard English?

Reviewer #1: Yes

Reviewer #2: Yes

6. Review Comments to the Author

Reviewer #1: understanding the need of the local community and better of the local community is a topic very few researcher would take up. The findings from the research would help the community to stand in its own legs with sustainability at its core.

Reviewer #2: After my suggestions, the author has done the necessary corrections so it can be accepted. I feel this is now suitable for publication.

7. PLOS authors have the option to publish the peer review history of their article (what does this mean?). If published, this will include your full peer review and any attached files.

Reviewer #1: **Yes: **Sriganeshvarun

Reviewer #2: **Yes: **Prof Shankar Chatterjee, India

---

## [Editor Report · Acceptance letter]

15 Jun 2023

PONE-D-22-32881R1 

Toward sustainable community-based tourism development: Perspectives from local people in Nhon Ly coastal community, Binh Dinh province, Vietnam 

Dear Dr. Nguyen:

I'm pleased to inform you that your manuscript has been deemed suitable for publication in PLOS ONE. Congratulations! Your manuscript is now with our production department. 

Kind regards, 

on behalf of

Dr. Sakul Kundra 

Academic Editor

PLOS ONE